# Numerical magnitude, rather than individual bias, explains spatial numerical association in newborn chicks

**Rosa Rugani[1,2]\*, Giorgio Vallortigara[3], Konstantinos Priftis[2], Lucia Regolin[1]**

[1]Department of General Psychology, University of Padova, Padova, Italy;
[2]Department of Psychology, University of Pennsylvania, Philadelphia, United States;
[3]Center for Mind/Brain Sciences, University of Trento, Rovereto, Italy

**Abstract** We associate small numbers with the left and large numbers with the right side of space. Recent evidence from human newborns and non-human animals has challenged the primary role assigned to culture, in determining this spatial numerical association (SNA). Nevertheless, the effect of individual spatial biases has not been considered in previous research. Here, we tested the effect of numerical magnitude in SNA and we controlled for itablendividual biases. We trained 3-day-old chicks (*Gallus gallus*) on a given numerical magnitude (5). Then chicks could choose between two identical, left or right, stimuli both representing either 2, 8, or 5 elements. We computed the percentage of Left-sided Choice (LC). Numerical magnitude, but not individual lateral bias, explained LC: LC2 *vs.* 2>LC5 *vs.* 5>LC8 *vs.* 8. These findings suggest that SNA originates from pre-linguistic precursors, and pave the way to the investigation of the neural correlates of the number space association.

## Introduction

Number knowledge and processing is fundamental for everyday living. A peculiar characteristic of numbers concerns their strong association with space. *Galton, 1880* first reported that humans, in many cases, describe and think of numbers as being represented on a mental number line (MNL) oriented from left to right. *Dehaene et al., 1993* provided seminal evidence for a left-to-right oriented MNL. Healthy humans are faster in processing small numbers through left-sided responses, and large numbers through right-sided responses. Traditionally, this effect, deemed SNARC (Spatial-Numerical Association of Response Codes), has been attributed to exposure to formal instruction, and therefore considered a by-product of culture, based on reading/writing conventions. Cultural aspects can in fact influence the orientation of the MNL. Arabs, who read from right to left, show an inverted SNARC effect (*Zebian, 2005*); people with mixed reading habits (e.g., Israelis) show no SNARC at all (*Shaki et al., 2009*). The placement of numbers along a left-to-right oriented MNL can be also modulated by the experimental context and adjusted by various forms of experience. Participants instructed to conceive numbers as distances along a ruler, showed a left-to-right oriented SNARC effect, whereas conceiving numbers as hours on a clock face elicited an inverted SNARC effect (*Bächtold et al., 1998*). Increasing evidence, however, has highlighted the importance of pre-linguistic and biologically-determined precursors of spatial-numerical associations (SNA). Evidence from infants rules out a primary influence of verbal counting in SNA orientation. Seven-month olds, habituated to left-to-right sequences of numerical magnitudes either increasing (e.g. 1-2-3) or decreasing (e.g. 3-2-1), at test looked longer at new increasing, but not decreasing, sequences. Whenever during habituation, the sequences were instead displayed from right-to-left, such bias was not reported for both increasing and decreasing sequences (*de Hevia et al., 2014*). The presentation of a small numerosity (two dots) or of a large numerosity (nine dots) oriented spatial attention of 8-month-old

**\*For correspondence:**
rosa.rugani@unipd.it

**Competing interests:** The authors declare that no competing interests exist.

**eLife digest** Most of the world modern-day languages are written from left to right – but what about numbers? As it turns out, the majority of people also represent numbers using a 'mental line', with smaller numbers on the left and larger numbers on the right. Some researchers argue that this phenomenon results from the way humans learn to read and write: in other words, that it is a by-product of culture, rather than an innate property of the brain.

Recent evidence suggests that newborn infants, as well as certain species of monkeys and birds, also associate smaller numbers with the left and larger numbers with the right side of space. This raises the possibility that human mental number line may stem from an ability that evolved before language, in a common ancestor of humans and other animals. Yet, critics claim that findings in infants and non-human species result from a failure to account for individual biases in responding.

To resolve this controversy, Rugani et al. trained three-day-old domestic chicks to approach a target board sporting five red squares. Chicks were then given the choice to approach two identical boards, which would both show two, five or eight red squares.

Rugani et al. showed that when both boards had two red squares, the chicks tended to approach the left-hand board more often than the right. By contrast, when both boards had eight red squares, the birds approached the right-hand board more often than the left. Importantly, no left-right bias was observed when the number of red squared remained unchanged (five). These results also could not be explained by individual chicks favoring the left or right side.

Instead, the findings suggest that even newborn animals tend to associate numbers with positions on a mental number line. Additional research is needed to determine the role of experience – or culture – in shaping this tendency, and future studies should also examine which brain regions support the association between number and space.

infants, respectively towards the left or the right side of space (*Bulf et al., 2015*). Nevertheless, these results could be determined by the interactions that few-months olds entertained with adults or their environment (*Patro et al., 2016*). SNA has been described even in 3-day-old newborns, strongly reducing the possible influence of the interaction with caregivers (*de Hevia et al., 2017*). Recently, *Di Giorgio et al., 2019* reported that newborns habituated to a numerical value (a group of 12 items), spontaneously associated a smaller number (four items) with the left side of space and a larger number (36 items) with the right side. Interestingly enough, the same number, for instance '12', was associated with the left side after habituation with a large number (36), but with the right side after habituation with a smaller number (4).

The studies that cast most doubts on the importance of language and symbolic thought for the origin of the SNA come from comparative research (*Brugger, 2015*; *Vallortigara, 2018*). Adult Clark's nutcrackers (*Rugani et al., 2010*) and rhesus monkeys *Drucker and Brannon, 2014* have shown unilateral, left-to-right oriented bias in associating numerosity with space. A spatial represen-tation of magnitude has been found also in gorillas and orangutans (*Gazes et al., 2017*). Though present in most apes, SNA is either left-to-right or right-to-left oriented, depending on the individ-ual. Idiosyncratic experiences, such as the interactions with caregivers, rather than differences related to species or handedness, are reported among the main factors that might determine the orientation of the SNA (*Gazes et al., 2017*). This interpretation makes it even clearer that the only way to rule out the role of culture as well as of caregiving experience is to test day-old (almost) *naive* animals. For instance, baby chicks (*Gallus gallus*) trained to respond to a target numerosity, sponta-neously associated a number smaller than the target with the left side of space, and a number larger than the target with the right side (*Rugani et al., 2015a*). This study strongly renovated the interest on the origin of the SNA (*Drucker and Brannon, 2014*; *Rugani and de Hevia, 2017*) and provided insights on testing SNA in non-verbal subjects. The paradigm of *Rugani et al., 2015a* has been applied onto different species, with mixed results. On one side, studies that merely applied the chicks' paradigm failed in finding a SNA (*Triki and Bshary, 2018*; *Beran et al., 2019*). On the other hand, studies that extrapolated the core idea and tuned the paradigm to the test situation did suc-cessfully find a SNA (*de Hevia et al., 2017*; *Di Giorgio et al., 2019*). An alternative explanation of the original study by *Rugani et al., 2015a* pointed at the importance of any, however subtle,

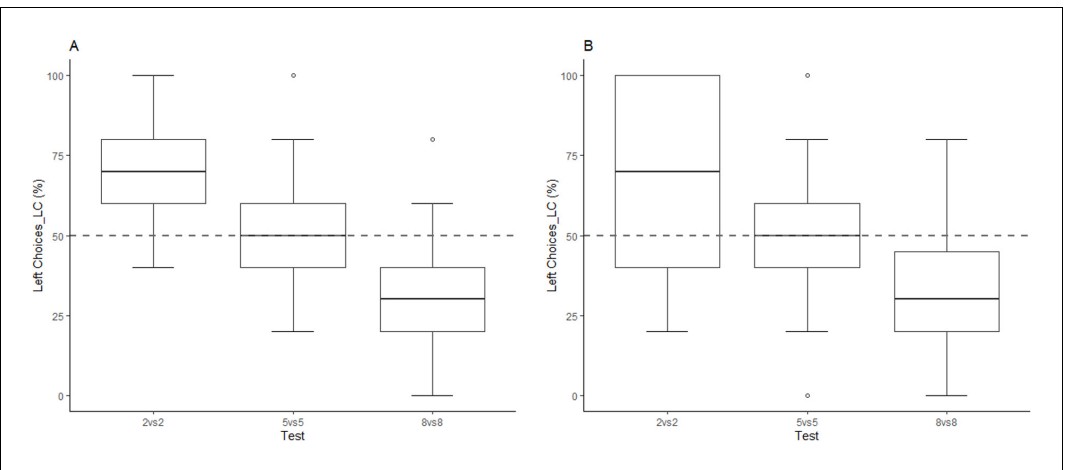

**Figure 1.** Left Choices (LC) as a function of numerical magnitudes. Left Choices (means, SE, 1Q and 3Q) in each test of Experiment 1 (**A**) and Experiment 2 (**B**). Dotted lines represent the chance level and dots represent the outliers, which were included in the final sample. Below, for each experiment we reported the Bayesian and the frequentist one-sample t-test *vs.* chance level. Experiment 1: Chicks took more left-sided choices when facing smaller magnitudes (two: BF >100; p<0.001, Cohen's d = 1.572), and more right-sided choices when facing larger magnitudes (eight: BF >100; p<0.001, Cohen's d = 1.505) than the one experienced during training (five); in the 5 *vs.* 5 test they did not show any bias (BF = 0.321; p=0.522, Cohen's d = 0.151). Experiment 2: Chicks took more left-sided choices when facing smaller magnitudes (two: BF = 51.417; p<0.001, Cohen's d = 1.072), and more right-sided choices when facing larger magnitudes (eight: BF = 23.070; p<0.001, Cohen's d = 0.979) than the one experienced during training (five); in the 5 *vs.* 5 test they did not show any bias (BF = 0.291; p=0.863, Cohen's d = 0.05).

individual biases either toward the left or the right which would be magnified over the course of the test (*Mangalam and Karve, 2015*), but see *Rugani et al., 2015b*). Furthermore, *Núñez and Fias, 2017* sustained that SNA in chicks could critically depend on having used novel stimuli at test with respect to the training. Chicks' responses might be triggered by novelty, rather than by numerosity. Therefore, *Núñez and Fias, 2017* highlighted the importance of testing the chicks on the same numerosity as that of the training (5 *vs.* 5).

The aim of the present study was to directly investigate the role of number magnitude *versus* individual spatial bias in 3-day-old domestic chicks.

At test, chicks were presented with the same numerosity already experienced at training (5). In different experiments this test was administered either as first or as last, to ascertain any role of experience in determining the orientation of the SNA. Data from the 5 *vs.* 5 control test were meant to be very insightful as no preference was expected according to the hypothesis that the bias is solely determined by the differences in numerical magnitude experienced by the animals. Hence, this test could unveil the presence of any idiosyncratic spatial bias. The performance of each chick in the 5 *vs.* 5 test was to this purpose employed to normalize performance scored in the other tests (2 *vs.* 2 and 8 *vs.* 8). This allowed to control for any individual bias as well as for responses guided by non-numerical variables (e.g., novelty). To test for lateralization in processing numerical magnitude, we scored the side from which chicks circumnavigated the panels. Chicks can use their eyes independently to process different visual stimuli; generally, they approached a stimulus from the side which allows an analysis with the eye connected to the most specialized hemisphere (*Daisley et al., 2009*). A left circumnavigation, which implies to look with the right eye, would indicate a preferential processing with the left hemisphere; a right circumnavigation, which implies to look with the left eye, would indicate a preferential processing with the right hemisphere.

Here, we trained 3-day-old chicks (n = 48) to circumnavigate a central panel depicting five elements to get a food reward. Then chicks underwent three consecutive tests, each consisting of five trials, in which two identical panels were presented. Each panel depicted either 2, 8 or 5 randomly arranged 1-cm-sided red squares. Each chick underwent a smaller number (2 *vs.* 2), a larger number

(8 *vs.* 8), and the control (5 *vs.* 5) test. On each test trial we scored the first inspected panel (left or right) and the side from which chicks circumnavigated the panel.

In Experiment 1, test presentation Order was: 2 *vs.* 2, 8 *vs.* 8, 5 *vs.* 5 (N = 12 chicks) or 8 *vs.* 8, 2 *vs.* 2, 5 *vs.* 5 (another sample of N = 12 chicks); chicks were randomly assigned to either group. In Experiment 2, the Order of the three tests was: 5 *vs.* 5, 2 *vs.* 2, 8 *vs.* 8 (N = 12 chicks) or 5 *vs.* 5, 8 *vs.* 8, 2 *vs.* 2 (another sample of N = 12 chicks); chicks were randomly assigned to either group. Sample size for each group was calculated, as indicated by the Ethical committee for animal welfare of the University of Padova, using the formula for quantitative variables: n=(2σ²)/(μ1-μ2)² x f(α,β); with the following values α = 0.05 e ß = 0.80 average = 70% standard deviation = 17%.

For each experiment, we calculated the percentage of trials in which the chick chose the left panel (Left Choices: LC). LC ranged from 0 (left panel never chosen) to 100 (left panel always chosen). Our main prediction was that Test (2 *vs.* 2; 8 *vs.* 8; 5 *vs.* 5) would affect LC; in particular we expected the following order restriction on 'Test' variable: LC(2 *vs.* 2)>LC(5 *vs.* 5)>LC(8 *vs.* 8). To assess whether chicks' behavior was based on number magnitude or on individual bias, for each chick we calculated a Small Number Bias (SNB): LC(5 *vs.* 5) – LC(2 *vs.* 2) and a Large Number Bias (LNB): LC(5 *vs.* 5) – LC(8 *vs.* 8). We expected a SNB <0, which indicates a left bias in responding to small magnitudes and a LNB >0, which indicates a right bias in responding to large magnitudes.

## Results

We conducted Bayes factor analyses using the version 0.9.12–4.2 of the Bayes Factor package in R, and using the default parameter values and JASP 0.11.1. We used the classification by *Lee and Wagenmakers, 2013* to interpret Bayes factor (BF). We conducted frequentist analyses using JASP 0.11.1.

### Experiment 1

We first considered the effect of Order and Test on the percentage of Left-sided Choices (LC; *Figure 1A*; *Source data 1*). Against the 'Intercept only' model, the Bayesian ANOVA(BfANOVA) produced an extreme evidence in favor of a Test effect (BF >100), (repeated measures Anova: F (2,44)=36.375; p<0.01, $\eta^2$ = 0.423); but no effect of Order (BF = 0.263), (repeated measures Anova: F(1,44) = 0.336; p=0.714, $\eta^2$ = 0.004), (*Source data 2*).

We then tested the equality constraints of our model by comparing the unconstrained model (LC_2 *vs.* 2 ≠ LC_8 *vs.* 8 ≠ LC_5 *vs.* 5) with every other possible somewhat constrained model (e.g.: LC_2 *vs.* 2 ≠ LC_8 *vs.* 8 = LC_5 *vs.* 5). The unconstrained model was preferred to all the possible constrained models by a factor ranging from 12 to >100. This is an evidence, ranging from strong to extreme, in favor of a differential performance in the three tests. To compare the order restrictions model (2 *vs.* 2 > 5 *vs.* 5 > 8 *vs.* 8) with the unconstrained full model (LC_2 *vs.* 2 ≠ LC_8 *vs.* 8 ≠ LC_5 *vs.* 5) we firstly ran a Markov Chain Monte Carlo (MCMC) which showed 9983/10000 cases consistent with our hypothesis. We found a moderate evidence in favor of the order restriction: LC_2 *vs.* 2>LC_5 *vs.* 5>LC_8 *vs.* 8 (BF = 5.990). Frequentist analyses confirmed that: LC_2 *vs.* 2 was significantly larger than chance (50%): mean = 71.667, SD = 19.486, t(23)=5.447, p<0.001, Cohen's d = 1.572; LC_8 vs. 8 was significantly smaller than chance: mean = 28.333, SD = 20.359, t(23) = −5.214, p<0.001, Cohen's d = 1.505; the LC_5 *vs.* 5 was not statistically different from chance: mean = 52.500, SD = 23.452, t(23) = 0.522, p=0.604, Cohen's d = 0.151, (*Source data 2*).

For what concerns the Number Bias, we firstly computed the SNB and the LNB, then we compared each Number Bias with the null = 0. T-test Bayes factor analysis yielded a very strong evidence in favor of the Number Bias for SNB, (BF = 49.037; One sample t test: t(23)=-3.922; p<0.001, Cohen's d = −0.801), (*Source data 3*) and an extreme evidence in favor of the Number Bias for LNB (BF >100; One sample t test: t(23)=4.872; p<0.001, Cohen's d = 0.994) (*Source data 4*). For what concerns the side of circumnavigation, we did not find any consistent evidence for each numerical magnitude: 2 *vs.* 2 (BF = 0.301; $X^2$ = 0.403; p=0.525, Phi = 0.058); 5 *vs.* 5 (BF = 3.314; $X^2$ = 5.444; p=0.020, Phi = 0.213); 8 *vs.* 8 (BF = 0.249; $X^2$ = 0.062; p=0.804, Phi = 0.023), see *Table 1*, Experiment 1; (*Source data 5*).

On the basis of these analyses we concluded that chicks' performance is affected by the number magnitude of elements faced at test.

**Table 1.** Data and results concerning the side of circumnavigation for each panel in all test conditions of both experiments (*Source data 5*).

| | | Experiment 1 | | | | | | | |
| | | Left panel | | Right panel | | BF | $X^2$ | P | Phi |
| Test | *Side* | *Left* | *Right* | *Left* | *Right* | | | | |
| 2 *vs.* 2 | Count | 51 | 35 | 18 | 16 | 0.301 | 0.403 | 0.525 | 0.058 |
| | % | 59.302 | 40.698 | 52.941 | 47.059 | | | | |
| 5 *vs.* 5 | Count | 41 | 22 | 25 | 32 | 3.314 | 5.444 | 0.020 | 0.213 |
| | % | 65.079 | 34.921 | 43.860 | 56.140 | | | | |
| 8 *vs.* 8 | Count | 21 | 13 | 51 | 35 | 0.249 | 0.062 | 0.804 | 0.023 |
| | % | 61.765 | 38.235 | 59.302 | 40.698 | | | | |
| | | Experiment 2 | | | | | | | |
| | | Left Panel | | Right Panel | | BF | $X^2$ | p | Phi |
| Test | *Side* | *Left* | *Right* | *Left* | *Right* | | | | |
| 2 *vs.* 2 | Count | 62 | 22 | 20 | 16 | 1.512 | 3.880 | 0.049 | 0.180 |
| | % | 73.810 | 26.190 | 55.556 | 44.444 | | | | |
| 5 *vs.* 5 | Count | 36 | 25 | 34 | 25 | 0.224 | 0.024 | 0.877 | 0.014 |
| | % | 58.333 | 41.667 | 57.626 | 42.373 | | | | |
| 8 *vs.* 8 | Count | 24 | 16 | 23 | 57 | 49.104 | 10.930 | <0.001 | 0.302 |
| | % | 60 | 40 | 28.750 | 71.250 | | | | |

## Experiment 2

We first considered the percentage of Left Choices (LC), separately for each test (*Figure 1B*; *Source data 6*). The BFANOVA against the 'Intercept only' model produced an extreme evidence in favor of a Test effect (BF = 1470.58; repeated measures Anova: F(2,44) = 16.736; p<0.001, $\eta^2$ = 0.277); but no effect of Order (BF = 0.249; repeated measures Anova: F(2,44)=4.388; p=0.018, $\eta^2$ = 0.073) (*Source data 7*). Frequentist analyses confirmed that: LC_5 *vs.* 5 did not statistically differ from chance: mean = 50.833, SD = 23.575, t(23) = 0.173, p=0.863, Cohen's d = 0.05; LC_2 *vs.* 2 was significantly larger than chance: mean = 70.000, SD = 26.375, t(23) = 3.715, p<0.001, Cohen's d = 1.072), and that LC_8 *vs.* 8 was significantly smaller than chance: mean = 33.333, SD = 24.077, t (23) = −3.391, p<0.001, Cohen's d = 0.979.

We then tested the equality constraints of our model by comparing the unconstrained model (LC_2 *vs.* 2 ≠ LC_8 *vs.* 8 ≠ LC_5 *vs.* 5) with every other possible somewhat constrained model (e.g.: LC_2 *vs.* 2 ≠ LC_8 *vs.* 8=LC_5 *vs.* 5). The unconstrained model was preferred to all the possible constrained models by a factor ranging from three to >100. This is an evidence ranging from moderate to extreme in favor of a differential performance in the three tests. To compare the order restrictions model (LC_2 *vs.* 2>LC_5 *vs.* 5>LC_8 *vs.* 8) with the unconstrained full model (2 *vs.* 2 ≠ 8 *vs.* 8 ≠ 5 *vs.* 5) we first ran a Markov Chain Monte Carlo (MCMC) which showed 9798/10000 cases consistent with our hypothesis. The BF of our order restriction model was 5.879 in favor of the restricted model against the full model, showing, thus, a moderate evidence in favor of the order restriction: LC2*vs.*2>LC5*vs.*5>LC8*vs.*8 (*Source data 7*).

For what concerns the Number Bias, we first computed the SNB and the LNB, then we compared each Number Bias with the null = 0. T-test Bayes factor analysis produce a moderate evidence for SNB (BF = 4.350 ± 0; One sample t test: t(23)=−2.752; p=0.011, Cohen's d = −0.562) (*Source data 8*) and a moderate evidence for LNB (BF = 3.525 ± 0; One sample t test: t(23)=2.64; p=0.015, Cohen's d = 0.539) (*Source data 9*). We did not find any consistent effect of the side of circumnavigation for 2 *vs.* 2 (BF = 1.512; $X^2$ = 3.880; p=0.049, Phi = 0.180) and 5 *vs.* 5 (BF = 0.224; $X^2$ = 0.024; p=0.877, Phi = 0.014). Nevertheless, there was an evidence for 8 *vs.* 8 (BF = 49.104; $X^2$ = 10.930; p<0.001, Phi = 0.302), see *Table 1*, Experiment 2 (*Source data 5*).

## Discussion

The results of Experiments 1 and 2 support the hypothesis that number magnitude affects chicks' performance. Interestingly, the numerical bias reported in Experiment two seems to be less strong than that reported in Experiment 1: the evidence in favor of the Number Bias was very strong-to-extreme in Experiment 1, but moderate in Experiment 2. Plausibly this reduced strength of the SNA is related to the first test, in which chicks experienced a magnitude identical to the training one. A recent paper showed that the effect of spatial-motor experience could modulate the SNA in pre-literate children. A short-term (≈15 min) spatial –and not numerical– training (i.e. playing a video game which required either left-to-right or right-to-left oriented movements) is sufficient to modulate the orientation of the spatial numerical association in 3- and 4-year-old children (*Patro et al., 2016*). Even if we did not find any consistent bias on the side of circumnavigation, side differences in circumnavigation seemed stronger in Experiment 2, where evidence of spatial numerical association were found to be somewhat weaker than in Experiment 1. This allows to speculate that experience may modulate the processing underlying numerical perception and its association with space.

On the contrary in Experiment 1, chicks experienced an appreciable variation in numerical magnitude in the very first test, in either direction: smaller (2) or larger (8). Such difference in the strength of the SNA opens new challenging opportunities to study the role of experience in modulating the SNA.

Taken together, our two experiments showed that number magnitude and space are strongly associated in young and naïve chicks. In presence of very limited numerical and environmental/spatial experience this association is left-to-right oriented. But is this effect sensitive to experience? Future studies are needed to assess whether and how the association of number and space, however predisposed, can be modulated by experience.

Chicks showed a left bias in the 2 *vs.* 2, a right bias in the 8 *vs.* 8 and no bias in the control test 5 *vs.* 5, irrespectively of testing order (*Figure 1A and B*). Our results cannot be explained by individual orienting biases or by preference for novelty (*Núñez and Fias, 2017*). Our results show, instead, that it is the relative numerical magnitude between the training and the testing values, which determines the direction of the bias. Moreover, the presence of a linear trend with three points of reference - LC(2 *vs.* 2)>LC(5 *vs.* 5)>LC(8 *vs.* 8)- poses further restrictions in favor of a spatial-numerical mapping guided by the mental number line (see *Núñez and Fias, 2017*).

One important issue is the adaptive significance, if any, of a directional number-space association for non-human (and human) animals. One could argue that the basic phenomena is the mapping of number to space, and that the ordering of such mapping is simply the outcome of chance processes. However, if this were the case, then one would expect different individuals showing different directional biases, for there would be no specific reason for an alignment in the direction of the space-number association in different individuals. Some more basic biological phenomenon may, however, be at work here. One hypothesis has been put forward by *Vallortigara, 2018*. It is based on evidence that the two sides of the brain provide qualitatively different contributions to the control of functions related to motivation and emotion - sometimes referred to as the 'valence hypothesis' (*Davidson, 2004*)- with the left and right sides of the brain specialized for positive (approach) or negative (withdrawal) aspects in the control of behavior. It is plausible that changes towards larger or smaller magnitudes are associated with prevalent activation of, respectively, the left (positive valence) and the right (negative valence) hemisphere, and, thus, that attention to contralateral hemispace arises from it. Even if not explicitly trained to such association, animals can establish that for appetitive stimuli like the ones used in our experiments, larger magnitudes are intrinsically better (and approachable) than smaller magnitudes. According to this hypothesis, when chicks are faced with either abrupt increase or decrease in numerosities this would evoke preferential activation of, respectively, the left hemisphere (positive emotion) or the right hemisphere (negative emotion). In turn, this would promote attending to the contralateral side of the activated hemisphere, i.e. to the left for changes from large to smaller numerosities and to the right for changes from small to larger numerosities. The hypothesis can be tested in future experiments by e.g. establishing specific associations between certain magnitudes and an aversive (rather than an appetitive) stimulus, thus inducing a reduction if not an inversion of the direction of the space–number association.

**Table 2.** Outline of the experimental procedures.

| Time | Procedures |
| --- | --- |
| Day 1, morning | Arrival and housing in standard conditions |
| Day 2, all day | Standard rearing conditions – no procedures |
| Day 3, from early morning to mid afternoon | Removal of Food jars (2 hr before shaping) |
| | Shaping – followed by 2 hr rest |
| | Training Session 1 |
| | Test 1 – followed by 1 hr rest |
| | Training Session 2 |
| | Test 2 – followed by 1 hr rest |
| | Training Session 3 |
| | Test 3 |
| | Social housing – end of procedures |

## Materials and methods

### Subjects

Subjects were forty-eight domestic chicks (*Gallus gallus*), Ross 308 Broiler (Aviagen). Twenty-four subjects took part in Experiment 1, and the other 24 took part in Experiment 2. We obtained chicks weekly from a local commercial hatchery (Agricola Berica, Montegalda, Vicenza, Italy).

All procedures chicks underwent are summarized in *Table 2*. On arrival, the chicks were a few-hours old. They were immediately housed in standard metal cages (28 cm × 32 cm × 40 cm), with food and water available ad libitum in transparent glass jars (5 cm in diameter, 5 cm high), and placed at the corners of the cages. Food and water were placed randomly, one jar per corner, and their position was changed daily. Each cage was illuminated by fluorescent lamps (36 W) located 45 cm above it. Temperature and humidity were constantly monitored and maintained respectively at 28–31°C and 68%. Twice a day we fed chicks with some mealworms (*Tenebrio molitor* larvae) as these were used as food reinforcement during training. We reared chicks in these conditions from the morning of arrival (from 11 a.m.) to the morning (8 a.m.) of Day 3, when the food jars were removed while water was left available. Two hours later (10 a.m.) birds underwent shaping, in which they learnt to circumnavigate a panel located in the center of the experimental apparatus. At the end of shaping each chick was placed back in its rearing cage and, two hours later, it underwent training. Immediately after the end of training the chick underwent Test 1. When the first test was over, the chick was placed back in its rearing cage for one hour, before entering a second session of training and, immediately after it, Test 2. At the end of Test 2, each chick was placed back in its rearing cage for about an hour, then the third training started, and, immediately thereafter, Test 3. At the end of all tests, chicks were caged in social groups of three birds, with food and water available ad libitum. A few hours later they were donated to local farmers.

### Apparatus

During training and test sessions we used the same experimental apparatus. This was located in a room adjacent to the rearing room. In the experimental room, temperature and humidity were controlled and maintained, respectively, at 25°C and 70%. Lighting was provided by four 58 W lamps, placed on the ceiling, 194 cm above the basement of the experimental apparatus.

The experimental apparatus consisted of a diamond-shaped arena (see *Figure 2*) made of uniformly white plastic panels. The external wall consisted of a 20 cm high plastic panel; the floor consisted of a white plastic sheet.

A transparent removable partition (10 cm × 20 cm) positioned at about 10 cm from the main vertex of the apparatus delimited the chick starting area. The transparent partition was used to confine the bird before the beginning of each training or testing trial. The chick was gently positioned and maintained in the starting area for five seconds, before being released within the arena. During this time the chick could access visually the inside of the arena and the panel(s). During the inter-trial

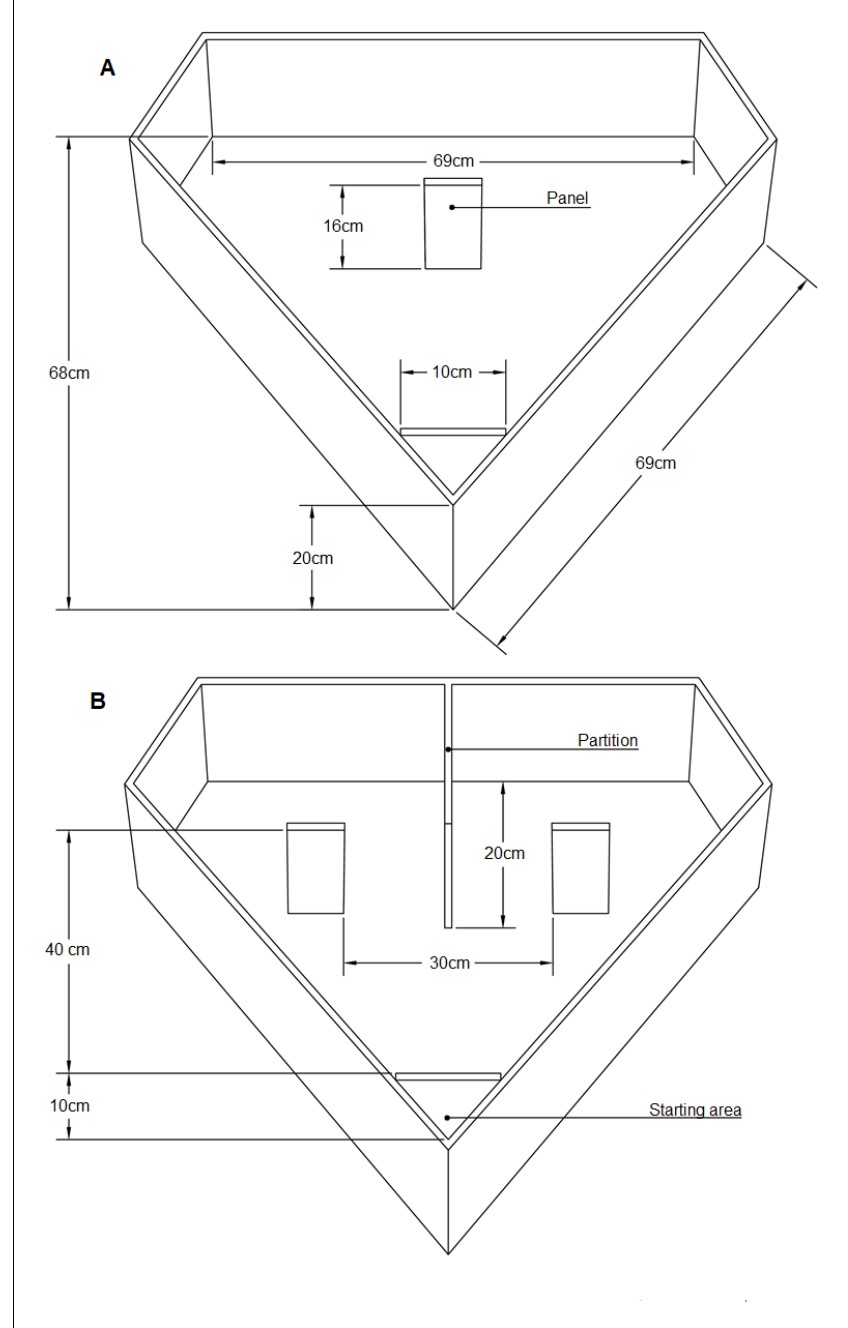

**Figure 2.** Experimental apparatus. Schematic representation of the apparatus used during training (**A**) and test (**B**).

period each chick was moved to a separate opaque box (20 cm ×40 cm × 40 cm) adjacent to the experimental apparatus, to prevent it from seeing the experimenter while cleaning the apparatus and changing the setup of the training/test stimuli. The stimuli were presented on panels (16 cm ×8 cm) provided with 3 cm sides bent back to prevent the chicks from spotting behind the panel (where the mealworm was hidden during training) before having walked around of it.

During training we used a single panel, located in the center of the apparatus, directly in front of the starting area and 40 cm away from it (see *Figure 2A*). During testing, we used two identical panels, spaced 30 cm apart, and located symmetrically one on the right side and one on the left side with respect to the main vertex (see *Figure 2B*). A partition, on the opposite side of the starting

area separated the far side of the apparatus in two symmetrical sectors, facilitating the scoring of chicks' choices.

## Stimuli

Training and test stimuli consisted of static 2D images. The stimuli depicted a number of red squares, printed on identical white rectangular boards (11.5 cm ×9 cm). On each trial a stimulus (during training) and a pair of stimuli (during testing) were placed on the panel(s).

The training stimuli depicted five red squares (1 cm ×1 cm). To prevent the chicks from learning to identify the stimuli on the basis of the spatial disposition of the squares, we created 20 different training stimuli (one for each training trials) differing one another for the spatial disposition of the squares on the board, which was randomly determined so that the distance between squares varied from 0.3 cm to 3.8 cm.

The test stimuli depicted either of 2, 5 or 8 identical red squares. Five different test stimuli, differing from one another in the spatial arrangement of the squares, were produced for the 2 *vs.* 2, 5 *vs.* 5, and 8 *vs.* 8 test. For each test stimulus we printed two identical copies. To the specific purpose of this study we designed the stimuli on the basis of those used in Experiment 1 of our previous study (*Rugani et al., 2015a*).

## Shaping and training

On the morning of Day 3 (i.e., the testing day) each chick underwent shaping, in which it was acquainted with feeding in the apparatus. A single panel was in place, and a mealworm was placed between the starting area and the panel. The chick was at first placed within the arena, in the starting area, without the confining partition, for a couple of minutes. During this time the chick was free to move around and get accustomed to the novel environment. In five subsequent trials we offered the chick a small mealworm (or a piece of a mealworm). In the first shaping trial, the mealworm was positioned closer to the starting area, while in the fifth shaping trial the mealworm was closer to the panel. Then chicks had to learn to search for food behind the panel. In this phase the chick was confined within the starting area. A plastic mealworm, which looked similar to a real one, was placed in front of the panel and then it was progressively moved (by a very fine thread handled and slowly dragged by the experimenter) behind the panel. Then the chick was released in the arena and could search for food, located behind the panel where an edible mealworm had been positioned. At the end of shaping, the chick confidently moved from the starting area and walked behind the screen to eat the reward.

Then chicks underwent training. On each training trial a stimulus was placed on the panel. The chick was confined in the starting area for five seconds and then it was released in the arena. The chick had one minute to circumnavigate the panel and to reach the reward. The training was over once the chick had circumnavigated the panel on 20 consecutive trials. In all training trials chicks received a food reinforcement. Previous studies, in which we used a procedure similar to this, have shown that after having found the food behind a panel depicting a certain number stimulus for a few times, the chicks learn to identify the panel by the number depicted on it (*Rugani et al., 2013*; *Rugani et al., 2014*). Overall, depending on the chick behavior, the training phase lasted from 10 to 20 min. Chicks that showed little interest in the food reinforcement (i.e., poor mealworm-following or eating behavior) in this phase, were discarded from the study: this occurred in about 25% of cases; these chicks are not included in the final sample. All chicks that completed the training phase moved on to the test session. Before the second and the third test, each chick underwent a training session identical to the first one.

## Test

This phase comprised three tests (2 *vs.* 2; 8 *vs.* 8; 5 *vs.* 5), each of them consisted of five trials. Test trials were never reinforced (i.e., chicks did not find any food reward behind the panels). On each test trial, chicks were firstly placed into the starting area, behind the transparent partition for about five seconds. Inside the arena, the two identical stimuli had been already positioned on the two new (left and right) panels, and were fully visible to the confined chick. Then the chick was released by lifting the transparent partition, and it was free to walk within the arena. As soon as the chick had circumnavigated one of the two panels, the trial was considered over. Only one choice was allowed

and was scored per trial. A choice was defined as when the head and at least ¾ of the chick's body had entered the area behind one of the two panels (beyond the side edges). At the end of each trial, the chick was moved in the opaque box outside of the apparatus, where it remained for about 15 s, during which time, the experimenter prepared the experimental setting for the following trial. On each trial the panels were shifted and the stimuli were changed. As soon as the new stimuli were in place, the chick was positioned back into the starting area, and the whole procedure was repeated.

If during a trial the chick did not choose one of the two panels within the available time (one minute), that trial was immediately repeated. The procedure continued until each chick had undergone three complete testing sessions of 5 valid trials each.

During the tests, subjects' behavior was observed from a screen connected to a video camera so as not to disturb the animals by direct observation, all trials were video-recorded.

For each test, we computed the number of trials in which each chick circumnavigated the left panel, and the percentages were computed as: (number of left choices/5)×100.

## Acknowledgements

This project has received funding from the European Union's Horizon 2020 research and innovation program under the Marie Sklodowska-Curie (grant agreement No. 795242 SNANeB) to RR; and a PRIN 2017 ERC-SH4–A (2017PSRHPZ) to LR and GV. This project has received also funding from the European Research Council (ERC) under the European Union's Horizon 2020 research and innovation program (grant agreement No. 833504 SPANUMBRA) to GV. This work was carried out within the scope of the project 'use-inspired basic research', for which the Department of General Psychology of the University of Padova has been recognized as 'Department of Excellence' by the Italian Ministry of University and Research.

## Additional information

### Funding

| Funder | Grant reference number | Author |
| --- | --- | --- |
| Horizon 2020 | 795242 SNANeB | Rosa Rugani |
| Horizon 2020 - Research and Innovation Framework Programme | 833504 SPANUMBRA | Giorgio Vallortigara |
| PRIN 2017 ERC-SH4-A | 2017PSRHPZ | Lucia Regolin |
| Ministry of Education, University and Research | Department of Excellence | Lucia Regolin |
| PRIN 2017 ERC-SH4-A | 2017PSRHPZ | Giorgio Vallortigara |

The funders had no role in study design, data collection and interpretation, or the decision to submit the work for publication.

### Author contributions

Rosa Rugani, Conceptualization, Data curation, Formal analysis, Funding acquisition, Investigation, Methodology, Writing - original draft, Project administration, Writing - review and editing; Giorgio Vallortigara, Conceptualization, Funding acquisition, Validation, Visualization, Writing - review and editing; Konstantinos Priftis, Conceptualization, Formal analysis, Supervision, Validation, Visualization, Writing - review and editing; Lucia Regolin, Conceptualization, Supervision, Funding acquisition, Validation, Visualization, Methodology, Writing - review and editing

### Author ORCIDs

Rosa Rugani  https://orcid.org/0000-0001-5294-6306
Giorgio Vallortigara  https://orcid.org/0000-0001-8192-9062
Konstantinos Priftis  https://orcid.org/0000-0002-5215-4621
Lucia Regolin  https://orcid.org/0000-0001-8960-0309

## Ethics

Animal experimentation: The experiments complied with all applicable national and European laws concerning the use of animals in research and were approved by the Italian Ministry of Health (permit number: 32662 granted on 19/07/2011). All procedures employed in the experiments included in this study were examined and approved by the Ethical Committee of the University of Padova (Comitato Etico di Ateneo per la Sperimentazione Animale - CEASA) as well as by the Italian National Institute of Health (NIH).

## Decision letter and Author response

Decision letter https://doi.org/10.7554/eLife.54662.sa1
Author response https://doi.org/10.7554/eLife.54662.sa2

# Additional files

### Supplementary files

• Source data 1. *Figure 1A*, Left Choices (LC) as a function of numerical magnitudes in Experiment 1.

• Source data 2. Left Choices (LC) and Order restriction: LC2*vs*.2>LC5 *vs*.5>LC8 *vs*.8 in Experiment 1.

• Source data 3. Small Number Bias (SNB) in Experiment 1.

• Source data 4. Large Number Bias (LNB) in Experiment 1.

• Source data 5. *Table 1*, Direction of Circumnavigation in Experiment one and in Experiment 2.

• Source data 6. *Figure 1B*, Left Choices (LC) as a function of numerical magnitudes in Experiment 2.

• Source data 7. Left Choices (LC) and Order restriction: LC2*vs*.2>LC5 *vs*.5>LC8 *vs*.8 in Experiment 2.

• Source data 8. Small Number Bias (SNB) in Experiment 2.

• Source data 9. Large Number Bias (LNB) in Experiment 2.

• Transparent reporting form

### Data availability

All data and codes have been uploaded as source data. The R codes used for data analyses, including the full list of parameters used have also been uploaded. Our datasets data are stored and backed up on the Research Data Unipd server (http://researchdata.cab.unipd.it/id/eprint/326). Research Data Unipd is a research data repository that meets the demands of FAIR (Findable, Accessible, Interoperable, Reusable) data storage, in accordance with the Guidelines on FAIR Data Management in Horizon 2020- July 2016.

The following dataset was generated:

| Author(s) | Year | Dataset title | Dataset URL | Database and Identifier |
|---|---|---|---|---|
| Rugani R, Vallortigara G, Priftis K, Regolin L | 2020 | Numerical magnitude, rather than individual bias, explains spatial numerical association in newborn chicks | http://doi.org/10.25430/researchdata.cab.unipd.it.00000326 | Research Data Unipd, 10.25430/researchdata.cab.unipd.it.00000326 |

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
