## [Decision Letter]

Thank you for submitting your article "Numerical magnitude, rather than individual bias, explains spatial numerical association in newborn chicks" for consideration by *eLife*. Your article has been reviewed by three peer reviewers, including Brian Butterworth as the invited Guest Editor and Reviewer #1, and the evaluation has been overseen by Christian Rutz as the Senior Editor.

The reviewers have discussed their reports with one another. While questions were initially raised about your article's suitability for *eLife*, given that it is essentially a response to critiques of an earlier publication, there was broad agreement that these well-conducted experiments provide strong results that usefully illuminate important debates in this field, and we have therefore decided to invite a revision.

Given that there were no disagreements between the reviewers on technical points, we have decided to append, on this occasion, their full separate reports below. The key points that require attention are: (1) simplification of the presentation of the statistical analyses; and (2) addition of some text attempting to offer an evolutionary explanation as to why chicks may prefer "small on the left".

Reviewer #1:

There are two views of the origin of SNA effect: task/cultura vs intrinsic to the relationship between magnitude and space. This is a very nice, well-conducted study testing a published critique by Nunez and Fias that a chick's orientation is determined by bias not magnitude. Here this critique is carefully tested using n vs n stimuli, with orientation shown to be determined by magnitude not bias. The majority of studies that have investigated this show a spatial-numerical association in a wide range of animals, including human infants. the authors conclude that SNA originates from pre-linguistic precursors. What is not clear is why chicks should behave in this way: what is the adaptive value of to the chick of a SNA with smaller magnitudes on the left?

Reviewer #2:

Rugani et al. tested 3 day old chicks in a spatial-numerical association task by first training them on 5 elements in the middle panel and then testing them with two identical number of elements (2, 5 or 8) presented to the left and right side of the choice box. Consistent with the conclusions of their earlier work, the comparison of the proportion of left choices revealed a SNARC-like effect; chicks were more likely associate left side of space with smaller numerosities whereas the right side of space with larger numerosities (based on choice behavior) suggesting that similar observations in humans are not due to enculturation.

Different from the earlier work, in the current study, numerosities were not presented repeatedly (e.g., compared to blocks of 5 trials) since the latter procedural variant of the task could lead to choice profiles resulting from the exaggeration of spatial biases coupled with shifts in spatial choices when the numerosity (presented in the earlier block of trials) is changed. Authors also included a test condition that was equal to the training numerosity (i.e., 5 vs. 5); this condition was specifically chosen to test for possible idiosyncratic spatial biases. In general, the study was conducted at high standards with better controls compared to the previous work of the same authors. I did not detect any objective errors in the paper. This being said, although these are very nice modifications done to the task specifically in light of Mangalam and Karve, 2015, I am not convinced that the novelty and added value of this empirical undertaking are high enough to be published in *eLife*. It seems to me like these experiments should have been parts of the previous work of the authors.

Reviewer #3:

This is a simple experiment on an important topic with very strong results. The strong and simple results are buried in a blizzard of highly technical statistical analyses that are in no way necessary or appropriate to the simple story these results tell. The authors may be congratulated for having done all these analyses but then told to eliminate most of them from their presentation.

---

## [Author Response]

Reviewer #1:There are two views of the origin of SNA effect: task/cultura vs intrinsic to the relationship between magnitude and space. This is a very nice, well-conducted study testing a published critique by Nunez and Fias that a chick's orientation is determined by bias not magnitude. Here this critique is carefully tested using n vs n stimuli, with orientation shown to be determined by magnitude not bias. The majority of studies that have investigated this show a spatial-numerical association in a wide range of animals, including human infants. the authors conclude that SNA originates from pre-linguistic precursors. What is not clear is why chicks should behave in this way: what is the adaptive value of to the chick of a SNA with smaller magnitudes on the left?

The issue of the adaptive value of such a directional SNA is indeed crucial, and we have now added in the Discussion a specific hypothesis in this regard. Please see the final paragraph of the Discussion section.